# The Evolving Landscape of Radiomics in Gliomas: Insights into Diagnosis, Prognosis, and Research Trends

**DOI:** 10.3390/cancers17091582

**Published:** 2025-05-06

**Authors:** Mehek Dedhia, Isabelle M. Germano

**Affiliations:** Department of Neurosurgery, Icahn School of Medicine at Mount Sinai, New York, NY 10029, USA; mehek.dedhia@icahn.mssm.edu

**Keywords:** radiomics, gliomas, brain tumors, neuro-oncology, machine learning

## Abstract

Gliomas are the most common kind of brain tumor that starts in the brain. Identifying gliomas early and figuring out exactly what kind they are is a challenge. It is also tough to tell how a patient will do over time, such as how well they will respond to treatment or how long they might live. Radiomics is a recent science that uses computer programs and artificial intelligence (AI) to study medical images, like MRIs or CT scans. It looks for patterns or tiny details in these pictures that the human eye might miss. AI helps by quickly and accurately analyzing a huge amount of image data. It can find clues in the pictures that tell doctors more about the tumor, including its type, how serious it is, and how it might react to treatment. This can help doctors choose the best treatment for each patient. If radiomics can be used to learn more about a person’s tumor from just an image, there might not be a need for risky surgery to obtain that information. It could also help predict what might happen to the patient so they and their doctors can make better plans. Scientists want to combine radiomics with other tools to make it even more powerful and helpful in the future.

## 1. Introduction

Gliomas are the most prevalent and aggressive form of primary brain tumors (PBTs) [1], arising from glial cells and comprising nearly 30% of all PBTs [2]. Their phenotypes range from lower-grade astrocytomas to the aggressive glioblastomas [3]. Despite advancements in research, gliomas present a significant clinical challenge due to their diffuse and infiltrative nature, which often leads to tumor recurrence. Recent progress in understanding the molecular and genetic diversity of gliomas has improved treatment options and facilitated individualized therapy at disease progression [4,5].

The first step in managing gliomas is making an early and accurate diagnosis. This is currently achieved using traditional imaging techniques such as magnetic resonance imaging (MRI), magnetic resonance spectroscopy (MRS), computed tomography (CT), and positron emission tomography (PET), analyzed by radiologists [6,7]. However, differentiating glioma subtypes or identifying subtler tumor characteristics is often difficult for the human eye. Additionally, interobserver variability and subjectivity can influence critical clinical decisions. Moreover, distinguishing between tumor progression, pseudo-progression, and treatment-related changes during treatment remains challenging [8,9,10,11], leading to uncertainties that impact clinical decisions.

An important aspect of diagnosis is identifying molecular signatures crucial for personalized treatment plans and targeted therapies. These include markers such as isocitrate dehydrogenase (IDH) mutation status [5], 1p/19q co-deletion [12], O^6^-methylguanine-DNA methyltransferase (MGMT) promoter methylation [13,14], and specific mutations such as the histone H3 gene, where the amino acid lysine at position 27 is replaced by methionine (H3K27M) [15], which are currently obtained through surgical biopsies. Non-invasive alternatives to tissue diagnosis offer significant clinical advantages, as biopsies may not always be feasible due to tumor location or patient comorbidities. Prognosis is also a key element in patient care, yet current assessments rely on clinical factors, histopathological grading, and genetic markers, which can be subjective and potentially inaccurate depending on the physician’s experience [16]. Machine learning techniques have already demonstrated promise in this field [17].

Radiomics—extracting quantitative data from medical images to identify patterns beyond human perception—presents a promising approach for gliomas [18,19]. It can extract large datasets from standard images (e.g., MRI, CT) that include shape, intensity, texture, and higher-order features. When combined with artificial intelligence (AI) and machine learning techniques, radiomics shows potential in aiding glioma diagnosis, predicting tumor molecular signatures without biopsy, and forecasting patient outcomes, as shown for other diseases [20,21]. Additionally, radiomics could enhance image interpretation with increased reproducibility and offer a non-invasive, patient-friendly alternative to surgical biopsy for upfront diagnosis and longitudinal monitoring.

Although radiomics has not yet been formally incorporated into clinical decision-making for brain tumors, research exploring its applications is rapidly growing. A comprehensive review of radiomics in brain tumors was published in 2023 [22], but a focused review on gliomas specifically is still lacking. This paper aims to fill that gap by systematically examining the current state of radiomics in gliomas, synthesizing recent findings, and offering a comprehensive perspective on its potential, limitations, and future directions in clinical care.

## 2. Materials and Methods

### 2.1. Literature Search Strategy and Data Extraction

A literature review was performed from the PubMed, Scopus, and Embase databases for the period 1 January 2023 to 31 December 2024 using the search strategy listed as follows: (RADIOMICS AND BRAIN TUMOR AND MACHINE LEARNING) OR (RADIOMICS AND BRAIN TUMOR HABITAT OR PERITUMORAL) OR (RADIOMICS AND PERITUMORAL AND MACHINE LEARNING AND BRAIN TUMOR).

Identified abstracts were included and imported to Covidence (Veritas Health Innovation, Melbourne, Australia) according to the inclusion and exclusion criteria listed as follows. Inclusion criteria: 1. Original articles on human subjects or technical reports, 2. clinical use of radiomics for gliomas, 3. full-text availability, and 4. English language manuscripts. Exclusion criteria: 1. Reviews, commentaries, or conference proceedings, and 2. case reports or articles with less than five patients.

Full-text articles were then reviewed, and data were extracted. This included the date of publication, number of subjects, features extracted, study topic within gliomas, glioma type studied, imaging sequences utilized, and genetic markers identified (for papers that studied the tumor molecular signature).

### 2.2. Quality Assessment

The quality of each study was also analyzed using the QUADAS-2 tool [23]. The criteria of the QUADAS-2 tool were modified to accommodate the lack of blinding that is seen in machine learning studies.

### 2.3. Statistical Analysis

Mean ± standard deviation (SD) or percentages were reported for continuous variables, and proportions were reported for categorical variables. A probability of (*p*) < 0.05 was considered significant. Statistical analysis was completed using R Studio v 4.1 (Boston, MA, USA).

## 3. Results

### 3.1. Radiomics Studies over Time

Of the initial 255 articles eligible for screening, 52 full-text papers were included in this review (Figure 1). These included 12,482 subjects in total. Over the study period, there was a steady number of publications without significant change over time. The article characteristics, including the number of subjects, tumor regions extracted, and performance metrics, are summarized in Table 1.

### 3.2. Radiomics Studies Characteristics

The three most common glioma radiomics study types were pertinent to the differential diagnosis of gliomas (26%), followed by the prediction of outcomes after diagnosis (21%), and the identification of features of the tumors (17%) (Figure 2).

Figure 3 depicts the distribution of articles by glioma type. The majority (56.9%) of papers discussed glioblastoma (GBM), followed by gliomas (35.3%) and brainstem/diffuse gliomas (7.8%). One paper discussed differentiating between radiation encephalopathy and glioma recurrence of brain tumors.

### 3.3. Image Sequences Utilized by Radiomics

Table 2 summarizes the MRI characteristics utilized and molecular signatures identified by the relevant studies. Three papers aimed to predict the expression of the IDH mutation and the MGMT mutation. Other molecular signatures that were identified were H3K27M Ki-67, a protein found in the cell nucleus that is associated with cell proliferation, widely used as a proliferation marker, as well as Vascular Endothelial Growth Factor (VEGF), chromosomes 1p/19q co-deletions, telomerase reverse transcriptase (TERT), and Alpha-Thalassemia/Mental Retardation Syndrome X-Linked (ATRX). A supplementary table demonstrates the MRI sequences used by the other studies and is divided by study type. Of note, one paper utilized CT segmentation to grade gliomas [68] and one paper utilized PET data to predict ratios of tumors to healthy brains and the corresponding GBM activity [74]. There were no significant differences across study types in how MRI characteristics were utilized. Of note, differently than radiogenomics, which combines radiomic features with genomic data, radiomics utilizes image features to identify molecular data. MRI sequences utilized in the other studies can be found in Appendix A.

### 3.4. Quality Assessment

Overall, most papers (90.4%) demonstrated a low risk of bias with respect to the reference standard. Thirty-nine papers (75%) had an unclear risk of bias with respect to patient selection, and all papers had a high bias with respect to the index test due to missing relevant details. There was also a low risk with respect to flow and timing. Patient selection risk was unclear in many studies due to a lack of information, and the risk of bias with respect to the index test was high due to the lack of a pre-specified threshold for the test. Applicability in all three domains had a relatively low concern (Figure 4).

## 4. Discussion

### 4.1. Radiomics Trends in Gliomas over a Decade

Radiomics, first introduced in 2012, gained traction quickly because of its potential to transform imaging from a qualitative to a quantitative science, leveraging advanced statistical and machine learning techniques to reveal hidden information within medical images [76]. A previous review [22], which broadly described the use of radiomics for all brain tumors, observed overall growth in the field, with a notable spike in 2021. However, it is challenging to determine how much of this increase can be attributed specifically to glioma-related studies. Our review demonstrates a steady increase in research over time regarding the use of radiomics for diagnosis and outcome prediction in glioma patients. This finding suggests that, while interest in the field remains strong, there may be signs of a plateau in research activity despite the continued evolution of this promising area.

### 4.2. Radiomics for Gliomas Upfront and Longitudinal Differential Diagnosis

Sensitivity and specificity are key metrics used to assess how well a model distinguishes between true positives and true negatives, respectively, for example, in the classification of gliomas. These metrics indicate the model’s ability to correctly identify different tissue types. To visually represent the model’s performance, graphical tools like the receiver operating characteristic (ROC) curve and the precision–recall curve are commonly employed. The ROC curve illustrates the trade-off between sensitivity and specificity across various classification thresholds [77]. In contrast, the precision–recall curve highlights the balance between precision and recall, especially when dealing with unbalanced datasets. Another widely used metric is the area under the ROC curve (AUC), where a higher AUC value signifies better performance in differentiating between positive and negative cases [78]. Additionally, the F1 score evaluates the balance between precision and recall, while a confusion matrix offers a comprehensive summary of the model’s overall performance.

Radiomics has shown strong promise in the field of differential diagnosis of gliomas. Our study shows that the AUCs of the studies range from 0.841 to 0.999, indicating a relatively high accuracy of diagnosis. While these utilize relatively small datasets, the AUC metrics indicate promise. The best-performing model, described by Wang et al. [25], utilized deep learning combined with advanced machine learning to differentiate meningiomas, gliomas, and pituitary tumors, which supports the utility of radiomics. However, the successful models that differentiated glioblastomas from metastases may have more clinical utility. These models also performed quite accurately as measured by the AUC curve and may have the potential to augment clinical practice. Furthermore, Lin et al. [37] described a model that aimed to distinguish between glioma recurrence and pseudo-progression, which can be more difficult to discern by the human eye. This model had an AUC value of 0.841. Such a value suggests that models such as this may have a role in aiding radiologists with confusing cases by suggesting the more likely diagnosis.

### 4.3. Can Radiomics Provide Patient-Specific Non-Invasive Tumor Molecular Signatures?

Patient-specific brain tumor molecular signatures refer to the unique genetic, epigenetic, and molecular characteristics present in an individual’s brain tumor. These signatures include mutations, gene expression profiles, DNA methylation patterns, and alterations in specific biomarkers, such as IDH mutations, 1p/19q co-deletion, and MGMT promoter methylation [79]. Identifying these molecular signatures is crucial because they provide insights into the tumor’s biology, including its aggressiveness, response to therapy, and potential for recurrence. Personalized treatment strategies can be developed by targeting these molecular abnormalities, improving the precision of therapy, and enhancing outcomes [80]. Furthermore, these signatures enable clinicians to predict prognosis more accurately, guide therapeutic decisions, and reduce unnecessary treatments, making them an essential component of precision medicine in brain tumor management.

Our study shows that radiomics has great potential for the identification of MGMT methylation with AUC values ranging from 0.66 to 0.923. Given that the lack of MGMT methylation is associated with higher temozolomide resistance in GBM patients [14], identifying this mutation early on in the disease course may help clinicians determine a more clinical beneficially treatment by weighing clinical efficacy versus known side effects of the drug. The study by Zhang et al. [56] demonstrated a model with an AUC of 0.88 that could be utilized to predict VEGF expression levels. This could augment clinical practice by aiding clinicians who may choose to include bevacizumab, a VEGF inhibitor, in their potential treatment plan. Additionally, with clinical trials in progress for vorasidenib, a mutant IDH1/IDH2 inhibitor, targeted therapeutics may soon be amended based on IDH status [5]. The three studies exploring the prediction of IDH1 mutation status [54,55,56] show AUC curves of 0.72 to 0.997. These results suggest that an IDH status likely requires some further refinement to achieve standardization. Finally, the prediction of chromosome 1p/19q co-deletion is necessary for the diagnosis and treatment of oligodendrogliomas [81].

### 4.4. Frequently Selected Radiomics Features

The models that aimed to extract feature/segment tumors also performed quite well, with AUC metrics ranging from 0.84 to 0.95. Such studies could accelerate the precise characterization of brain tumors and the identification of peritumoral edema and invasion. Timely and accurate delineation of these regions may help clinicians determine the tumor’s aggression and treatment effects and guide prognosis. Additionally, such models could be longitudinally integrated into the clinical workflow to provide a before and after treatment comparison of imaging. This could help clinicians better understand if the therapy is having its intended effect or if a modification in the treatment plan is required. Furthermore, distinguishing between tumor regions and areas of edema can provide crucial information for planning surgical resection and targeted radiotherapy during the treatment course. Precise segmentation of the tumor will aid surgical planning by ensuring that healthy tissue is preserved as much as possible. Visualization of edema may influence preoperative decisions like the administration of steroids and help predict clinical outcomes [82,83]. With advances in technology, these models may even provide utility in intraoperative imaging, providing continuous feedback to surgeons.

### 4.5. Current Gaps and Future Opportunities in Radiomics

It is also important to note that between papers, there was likely a lack of standardization in how the images were acquired, given that these images were acquired at different institutions with different technicians and machines. Minor variations in the machines used and how they are used can lead to subtle differences in images, negatively impact the generalizability of each model, and lead to model overfitting. This prevents a study’s findings from being extrapolated to other centers and underscores the idea that these models must be tested on large, multi-institutional cohorts to further validate their findings.

Models that aimed to predict patient prognosis had more limited utility, with the AUC metrics ranging from 0.7 to 0.8. However, prediction of patient prognosis is highly complex and involves many other factors that cannot be captured in imaging, such as patient comorbidities, age, treatment, and other characteristics. As a result, including such factors in the models may help improve accuracy so that they can reach clinical utility. This may involve the integration of clinical data, such as in the form of electronic health records. None of the studies mentioned in this review included the patient’s clinical data. The inclusion of clinical data, which can also be performed longitudinally, could support a more dynamic treatment approach and prognosis prediction by inputting real-time data into the model. This would allow clinicians to make better-informed decisions on a daily basis throughout the treatment course.

The evolving landscape of glioma treatment has prompted the development of new therapeutic concepts, including radionuclide therapy guided by molecular imaging. This process, known as theranostics, is based on a combined approach that integrates therapy and diagnosis, offering a potential new role for radiomics in addressing the clinical needs of this patient population. Targeted radionuclide therapies have demonstrated clinical relevance in extracranial tumor types [84] and hold promise for application in both primary and metastatic brain tumors.

### 4.6. Study Limitations

This review has limitations worth mentioning. First, despite optimizing our search terms, it is likely that some relevant papers were missed due to the limited scope of our search. However, given the diversity of included studies, we believe we have captured the majority of relevant papers, which should be representative of the broader body of work. Additionally, the lack of blinding in the paper review may introduce bias, though we mitigated this by having two authors independently review each paper. Many of the included studies also show a high potential for bias due to the absence of blinding and preset thresholds. Given the unique nature of these studies, it may be necessary to develop a modified tool for assessing bias. Lastly, although the models demonstrated strong AUC performance, all studies relied on retrospective data. To fully assess their clinical applicability, these models should be tested in prospective clinical registries. Finally, although over 12,000 patients were included in this review, each study itself was relatively small, further hindering its generalizability. There is an opportunity to further the strength of these models through collaboration and the utilization of shared datasets. This would allow the models to be trained on a larger collection of data and enhance the models’ predictive capabilities.

## 5. Conclusions

In conclusion, radiomics holds significant potential for the diagnosis and prognosis of glioma patients. It could enhance clinical practice by enabling faster diagnoses and guiding targeted therapies, both of which may improve patient outcomes. For broader implementation, models must be trained and validated on larger, multi-institutional datasets to ensure their accuracy and generalizability. Additionally, radiomics has not yet achieved sufficient clinical utility for reliable prognosis prediction. Future research focusing on integrating radiomics with other modalities, such as genomics and liquid biopsy techniques, is needed to improve its predictive capabilities.

## Figures and Tables

**Figure 1 cancers-17-01582-f001:**
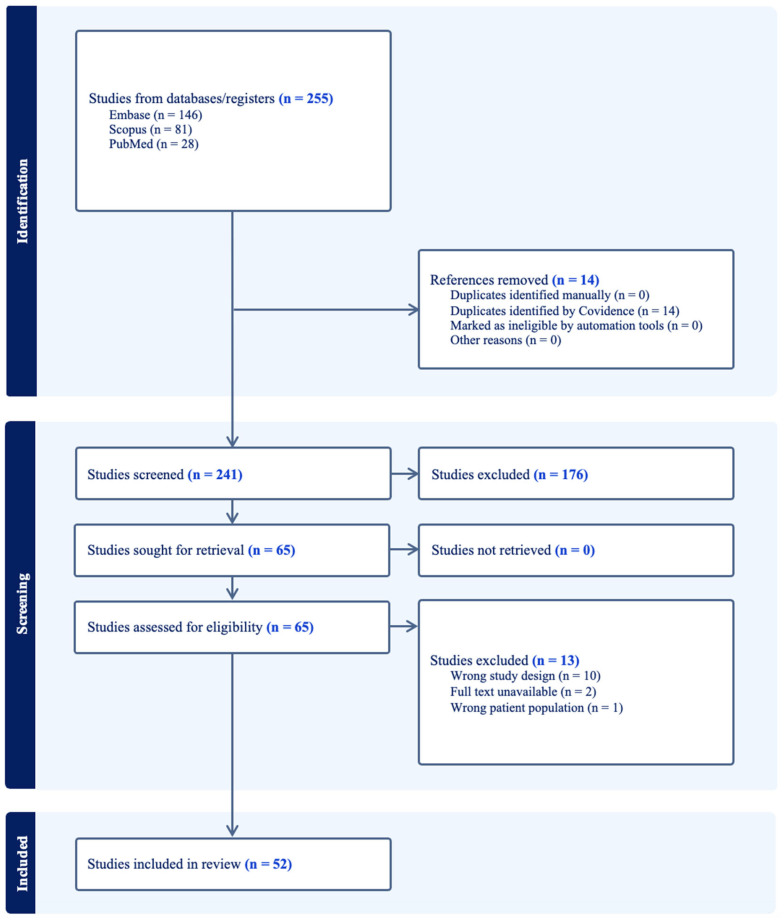
Prisma flow chart. This is a chart demonstrating how many studies were excluded and at what stage.

**Figure 2 cancers-17-01582-f002:**
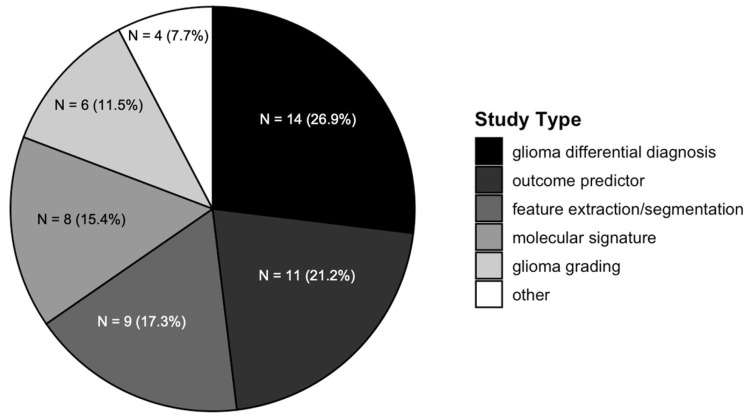
Glioma radiomics publications by study type. This is a pie chart demonstrating the distribution of publications by study type.

**Figure 3 cancers-17-01582-f003:**
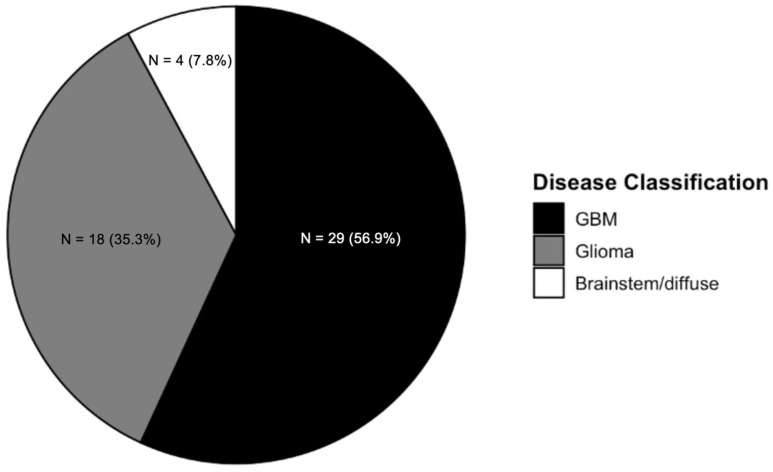
Glioma radiomics publications by disease classification. This is a pie chart demonstrating the distribution of publications by disease classification.

**Figure 4 cancers-17-01582-f004:**
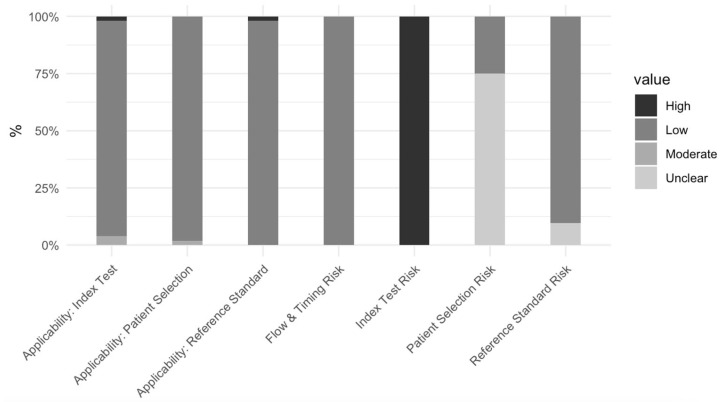
Bar graph summarizing QUADAS-2 distribution. This is a bar graph that shows the distribution of papers based on different metrics in the QUADAS-2 tool.

**Table 1 cancers-17-01582-t001:** Study characteristics presented by study type. This table lists all the included studies categorized by study topic and their characteristics.

*Study*	*N*	*Features Extracted*	*Validation Method*	*Performance*
**1.** **Differential Diagnosis**				
A brain tumor computer-aided diagnosis method with automatic lesion segmentation and ensemble decision strategy [24]	1022	ET, NCR, PTE	10 fold	AUC = 0.9
A hybrid deep learning scheme for MRI-based preliminary multiclassification diagnosis of primary brain tumors [25]	66	Tumor	5 fold	AUC = 0.9
AI-based classification of three common malignant tumors in neuro-oncology: A multi-institutional comparison of machine learning and deep learning methods [26]	208	ET, NCR, PTE	5 fold	AUC = 0.919
An explainable MRI-radiomic quantum neural network to differentiate between large brain metastases and high-grade glioma using quantum annealing for feature selection [27]	72	PTE, tumor	5 fold	AUC = 0.95
Differential diagnosis of radiation encephalopathy and post-radiation brain tumor recurrence by machine learning models based on contrast-enhanced MRI [28]	67	PTE, Tumor	3 fold	AUC = 0.9805
Diffusion-weighted imaging and arterial spin-labeling radiomics features may improve differentiation between radiation-induced brain injury and glioma recurrence [29]	66	Solid tumor	10 fold	AUC = 0.96
Discrimination between glioblastoma and solitary brain metastasis using conventional MRI and diffusion-weighted imaging based on a deep learning algorithm [30]	123	PTE, tumor	No CV	AUC = 0.956
Evaluating autoencoders for dimensionality reduction of MRI-derived radiomics and classification of malignant brain tumors [31]	93	Whole tumor, ET, NCR, PTE	5 × 5 fold	AUC = 0.91
Glioblastoma and solitary brain metastasis: differentiation by integrating demographic MRI and deep learning radiomics signatures [32]	115	ET, PTE	No CV	AUC = 0.999
Graph-radiomics learning (GrRAiL): Application to distinguishing glioblastoma recurrence from pseudo-progression on structural MRI [33]	106	ET, NCR, PTE	CV	AUC = 0.85
High-performance presurgical differentiation of glioblastoma and metastasis by means of multiparametric neurite orientation dispersion and density imaging (NODDI) radiomics [34]	57	NEC, solid tumor, PTE	5 fold	AUC = 0.701 (nec), 0.820 (PTE), 0.904 (whole tumor)
MRI characteristics of H3 G34 mutant diffuse hemispheric gliomas and possible differentiation from IDH wild-type glioblastomas in adolescents and young adults [35]	53	ET, NET, PTE	5 fold	AUC = 0.925
Novel 3D magnetic resonance fingerprinting radiomics in adult brain tumors: a feasibility study [36]	33	ET, NET, PTE	No CV	AUC = 0.85
Radiomic features on multiparametric MRI for differentiating pseudo-progression from recurrence in high-grade gliomas [37]	109	Whole tumor, PTE	10 fold	AUC = 0.841
**2.** **Outcome Predictors**				
Automated neural network-based survival prediction of glioblastoma patients using preoperative MRI and clinical data [38]	369	NCR, NET, ED, ET	5 fold	51.7% accuracy
Brain tumor segmentation and survival prognostication using 3D radiomics features and machine learning algorithms [39]	325	Tumor, PTR	No CV	73.8% accuracy
Clinical and magnetic resonance imaging radiomics-based survival prediction in glioblastoma using multiparametric magnetic resonance imaging [40]	93	Tumor, ED	5 fold	C index = 0.7
Cortical myelin and thickness mapping provide insights into whole-brain tumor burden in diffuse midline glioma [41]	154	Whole tumor	No CV	AUC = 0.84
Deep learning of time–signal intensity curves from dynamic susceptibility contrast imaging enables tissue labeling and prediction of survival in glioblastoma [42]	272	NET, ET	10 fold	C index = 0.72
Early prognostication of overall survival for pediatric diffuse midline gliomas using MRI radiomics and machine learning: A two-center study [43]	69	Whole tumor, tumor core	5 fold	77% accuracy
Fully automated radiomics-based machine learning models for multiclass classification of single brain tumors: glioblastoma, lymphoma, and metastasis [44]	401	ET, NET	10 fold	AUC = 0.878
Magnetic resonance imaging (MRI)-based intratumoral and peritumoral radiomics for prognosis prediction in glioma patients [45]	163	ET, NET, PTE	5 fold	AUC = 0.91
Overall survival prediction from brain MRI in glioblastoma [46]	285	ET, NET, PTE	2 fold	82% accuracy
Radiomics-based machine learning with natural gradient boosting for continuous survival prediction in glioblastoma [47]	865	ET, NCR	No CV	AUC = 0.791
Time-to-event overall survival prediction in glioblastoma multiforme patients using magnetic resonance imaging radiomics [48]	119	Core tumor, ET, NCR	3 fold	C index = 0.77
**3.** **Molecular Signatures**				
Assessment of MGMT promoter methylation status in glioblastoma using deep learning features from multi-sequence MRI of intratumoral and peritumoral regions [49]	356	NCR, ET, PTE	5 fold	AUC = 0.923
Classification of 1p/19q status in low-grade gliomas: experiments with radiomic features and ensemble-based machine learning methods [50]	159	Whole tumor	No CV	AUC = 0.846
Combined evaluation of T1 and diffusion MRI improves the non-invasive prediction of H3K27M mutation in brainstem gliomas [51]	126	Whole tumor	10 fold	AUC = 0.9246
Diffusion MRI-based connectomics features improve the non-invasive prediction of H3K27M mutation in brainstem gliomas [52]	133	Whole tumor	10 fold	AUC = 0.9136
Fused deep learning paradigm for the prediction of o6-methylguanine-DNA methyltransferase genotype in glioblastoma patients: A neuro-oncological investigation [53]	585	PTE, Tumor core, ET	5 fold	AUC = 0.753
Radiomic features of contralateral and ipsilateral hemispheres for prediction of glioma genetic markers [54]	143	ET, NET, PTE	5 fold	IDH AUC = 0.72
The application value of the support vector machine model based on multimodal MRI in predicting IDH-1mutation and Ki-67 expression in glioma [55]	309	Solid tumor	10 fold	IDHAUC = 0.997 KI67AUC = 0.965
Whole-brain morphologic features improve the predictive accuracy of IDH status and VEGF expression levels in gliomas [56]	182	ED, ET, NCR	10 fold	AUC = 0.88
**4.** **Feature Extraction/Segmentation**				
Auto-segmentation and classification of glioma tumors with the goals of treatment response assessment using deep learning based on magnetic resonance imaging [57]	285	Whole tumor, ET, NEC, PTE	No CV	99.1% accuracy
Deep learning automatic semantic segmentation of glioblastoma multiforme regions on multimodal magnetic resonance images [58]	1251	ET, PTE, NET, whole tumor	No CV	Precision = 9.3
Distinguishing tumor cell infiltration and vasogenic edema in the peritumoral region of glioblastoma at the voxel level via conventional MRI sequences [59]	28	NET, PTE	5 fold	AUC = 0.93
Evaluating the relationship between magnetic resonance image quality metrics and deep learning-based segmentation accuracy of brain tumors [60]	306	ET, ED, NCR	5 fold	Dicescore = 0.7280,
Functional and structural reorganization in brain tumors: a machine learning approach using desynchronized functional oscillations [61]	11	ET, NET, PTE, NCR	19 fold	Correlation = 0.795
Identification of radiomic signatures in brain MRI sequences T1 and T2 that differentiate tumor regions of midline gliomas with H3.3K27M mutation [62]	12	Tumor, PTE	No CV	5% ofcharacteristic
Quantification of radiomics features of peritumoral vasogenic edema extracted from fluid-attenuated inversion recovery images in glioblastoma and isolated brain metastasis, using T1-dynamic contrast-enhanced perfusion analysis [63]	48	NET, PTE	No CV	AUC = 0.84
Radiomics-based evaluation and possible characterization of dynamic contrast-enhanced (DCE) perfusion derived different sub-regions of glioblastoma [64]	89	ET, NET, ED, NCR	No CV	AUC = 0.89
Training and comparison of nnU-Net and deepmedic methods for auto-segmentation of pediatric brain tumors [65]	339	ET, NET, PTE, NCR	5 fold	Dicescore = 0.9
**5.** **Glioma Grading**				
Deriving quantitative information from multiparametric MRI via radiomics: evaluation of the robustness and predictive value of radiomic features in the discrimination of low-grade versus high-grade gliomas with machine learning [66]	158	ET, NET, ED	5 fold	AUC = 0.92
Glioma subtype prediction based on radiomics of tumor and peritumoral edema under automatic segmentation [67]	424	Tumor, PTE	5 fold	AUC = 0.945
Grading of gliomas by contrast-enhanced CT radiomics features [68]	62	Whole tumor	5 fold	AUC = 0.98
Machine learning-empowered brain tumor segmentation and grading model for lifetime prediction [69]	369	NCR, NET, ED, ET	No CV	98% accuracy
Radiomics analysis of quantitative maps from synthetic MRI for predicting grades and molecular subtypes of diffuse gliomas [70]	124	ET, NET, PTE	10 fold	AUC = 0.92
Use of radiomics models in preoperative grading of cerebral gliomas and comparison with three-dimensional arterial spin labeling [71]	105	Whole tumor	No CV	AUC = 0.929
**Other**				
Deep learning automates bidimensional and volumetric tumor burden measurement from MRI in pre- and post-operative glioblastoma patients [72]	1264	ET, PTE	5 fold	Coefficient = 0.959
Predicting peritumoral glioblastoma infiltration and subsequent recurrence using deep-learning-based analysis of multiparametric magnetic resonance imaging [73]	229	NET, ET, PTE	10 fold	OR = 6.90 to 12.63
Radiomics in determining tumor-to-normal brain SUV ratio based on11C-Methionine PET/CT in glioblastoma [74]	40	Whole tumor	No CV	Spearman = 0.58
The assessment of glioblastoma metabolic activity via 11C-Methionine PET and radiomics [75]	40	active region	No CV	Spearman = 0.7
**Total N= 12,482**	12,482			

Abbreviations: ET = enhancing tumor; NET = non-enhancing tumor; ED = edema; NCR = necrotic region; PTE = peritumoral edema; CV = cross-validation.

**Table 2 cancers-17-01582-t002:** MRI sequences utilized for glioma molecular signature studies. This is a table demonstrating which MRI sequences were utilized by the papers that discussed molecular signature prediction, and which markers were discussed.

Author	Year	T1w	T1wCE	T2w	FLAIR	DTI	Molecular Signature
Molecular Signature (n = 10)
Liang et al. [55]	2024	1	1	1	1	1	IDH, Ki-67
Yang et al. [51]	2024	1	0	0	0	1	H3K27M
Yu et al. [49]	2024	1	1	0	0	0	MGMT
Zhang et al. [56]	2024	1	1	1	1	0	IDH, VEGF
Medeiros et al. [50]	2023	0	0	1	0	0	1p/19q
Saxena et al. [53]	2023	1	1	1	1	0	MGMT
Wang et al. [54]	2023	1	1	1	1	0	IDH, MGMT, TERT, and ATRX
Yang et al. [52]	2023	1	1	1	0	1	H3K27M
Total	8	8	8	5	4

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
