# Peer review of "The Evolving Landscape of Radiomics in Gliomas: Insights into Diagnosis, Prognosis, and Research Trends"

_cancers, 2025, doi:10.3390/cancers17091582_

Round 1

Reviewer 1 Report

Comments and Suggestions for Authors

The aim of the review, as stated in the title, is to present trends in diagnosis, prognosis, and research within evolving landscape of radiomics in gliomas. In that aspect the ms doesn't fullfills the assumption. It is just a collection of the references (properly searched, sure) with their general analysis. The ms searched covered 2 years - too short time period to show "evolution", gives little relevant information about different kinds of gliomas (very heterogenous group), and actuallu no information of solely radiomics achievements in the field of diagnosis and prognosis (markers?). It also underestimates the information from molecular field, which, for now, are stricltly bind to treatment and survival. The ms does not give any cross-sectional summary of existing radiomic markres connected with certain glioma types.

My minor comments are as follows:

  1. The review concentrates on radiomics. However in many points, starting in the Abstract and SS, it refers to molecular data, as well as treatment response prognostication and survival prognosis. As for now, the latter clinical aspects are correlated with molecular data, radiomics itself works still in correlation with molecular ones. That should be clearly stated where relevant, because then the "name" changes to radiogenomics, definitely in paragraph 3.3.
  2. Simple Summary should be reformulated (according to Editor's requirements) not being just a shorten Abstract's version. Eg. For a layman it should be stated why gliomas are the problem, ahat is radiomics, and how AI helps.

  3. Table 1 - no references to the studies; should be better visually presented - maybe horizontally?

  4.  

    the Figures and Tables descriptions ahould be edited according to Editor's requirements.

  5. The reference to tables and figures in the text shouldn't be in bold.

Author Response

  1. The review concentrates on radiomics. However in many points, starting in the Abstract and SS, it refers to molecular data, as well as treatment response prognostication and survival prognosis. As for now, the latter clinical aspects are correlated with molecular data, radiomics itself works still in correlation with molecular ones. That should be clearly stated where relevant, because then the "name" changes to radiogenomics, definitely in paragraph 3.3.

We thank the reviewer for this comment. Differently than radiogenomics, which combines radiomic features with genomic data, radiomics utilizes image features to identify molecular data. Therefore, we do not see the need to change the nomenclature. However, we clarify this important point in paragraph 3.3 on page 10, line 156. 

  1. Simple Summary should be reformulated (according to Editor's requirements) not being just a shorten Abstract's version. Eg. For a layman it should be stated why gliomas are the problem, what is radiomics, and how AI helps.

Thank you for this suggestion. The simple summary was re-written following the reviewer’s comment and is on page 1, lines 10-21. 

  1. Table 1 - no references to the studies; should be better visually presented - maybe horizontally?

Thank you for this suggestion. The references were previously provided in the Supplementary Table 1. In the revised manuscript, they are provided within Table 1 and listed in the paper reference list. Table 1 has been has been changed to be horizontally oriented.  

  1. the Figures and Tables descriptions should be edited according to Editor's requirements

Thank you for this comment. Figure and table captions have been added.

  1. The reference to tables and figures in the text shouldn't be in bold.

Thank you for this comment. This has been corrected in text.

Reviewer 2 Report

Comments and Suggestions for Authors

This review highlights the promising role of radiomics in improving the diagnosis, molecular characterization, and outcome prediction of gliomas, offering a non-invasive path toward personalized and longitudinal care. Future integration with multi-modal data and large-scale studies is essential for clinical translation. The manuscript presents a timely and well-structured review of radiomics in glioma management. Minor revisions are recommended to improve clarity and impact.
1.    Please provide the sources and corresponding references for the data presented in Table 1 to ensure transparency.
2.    While the manuscript comprehensively reviews the application of radiomics in diagnosis and prognosis, it overlooks its emerging role in treatment planning and response assessment. The authors are encouraged to include a discussion on the integrative role of radiomics in theranostics, supported by recent studies.
3.    Several references cited are relatively outdated. Please incorporate the latest literature to reflect the current state of research and enhance the relevance of the review.

Author Response

  1. Please provide the sources and corresponding references for the data presented in Table 1 to ensure transparency.

Thank you for this suggestion. These were previously provided in the Supplementary Table 1. They are now cited in Table 1 and listed within the main paper reference list.

  1. While the manuscript comprehensively reviews the application of radiomics in diagnosis and prognosis, it overlooks its emerging role in treatment planning and response assessment. The authors are encouraged to include a discussion on the integrative role of radiomics in theranostics, supported by recent studies.

We thank the reviewer for this comment. This is now included in the discussion, paragraph 4.5, lines 274 - 280.

  1. Several references cited are relatively outdated. Please incorporate the latest literature to reflect the current state of research and enhance the relevance of the review

Thank you for this comment. We have updated the reference list to ensure all listed references are published within the last 10 years with the exception of the following for the reason highlighted: reference 14 is a landmark study with 100s of previous citations; reference 23 is the original paper describing the QUADAS tool; reference 75 is the one of the first papers to mention radiomics; references 76 and 77 are methods papers; reference 80 is a landmark study with 100s of previous citations.